environmental chemistry/organic chemistry

algae biomass, Py-GC/MS, pyrolysis, catalyst, HZSM-5/MCM-41composite molecular sieve

**Author for correspondence:**
Zhaoping Zhong
e-mail: zzhong@seu.edu.cn

# Research on catalytic pyrolysis of algae based on Py-GC/MS

## Hao Zhao, Zhaoping Zhong, Zhaoying Li and Wei Wang

Key Laboratory of Energy Thermal Conversion and Control of Ministry of Education, School of Energy and Environment, Southeast University, Nanjing 210096, People's Republic of China

HZ, 0000-0003-4016-1311

In order to improve the quality of catalysis products of algae, composite molecular sieve catalyst was prepared by digestion and crystallization of HZSM-5 to reduce the oxygen content of the catalytic products. According to the analysis of the pyrolysis products, the best preparation conditions were chosen of tetra propylammonium hydroxide (TPAOH) solution $2.0 \, \text{mol} \, l^{-1}$, cetyltrimethylammonium bromide (CTAB) solution 10 wt%, crystallization temperature 110°C, digestion–crystallization time: 24–24 h. The results indicate that the main function of catalysts is to promote the conversion of alcohols into hydrocarbons by reducing energy barriers. Catalysed by the composite molecular sieve, the content of alcohols in the pyrolysis products decreased from more than 30% to less than 10%, the content of hydrocarbons increased from 20% to nearly 60%, while all the adverse components remained at a low level, which indicates that the catalytic pyrolysis products are of high quality. The great deoxidation effect of composite molecular sieves is not only due to the expansion of the range of organic matter during re-pyrolysis, but also the increasing of the residence time of pyrolysis products inside the structure for the external mesoporous structure.

## 1. Introduction

The global warming caused by fossil energy [1,2] has attracted worldwide attention, and as non-renewable energy, it is difficult for fossil energy to support the sustainable development of society in the future. The development of renewable energy has become a worldwide trend, such as solar, wind, tidal and biomass energy. According to relevant research [3], biomass energy, a kind of clean energy, plays an important role in alleviating environment pollution problems. The utilization of biomass could be realized by means of biomass energy conversion technology which mainly includes liquefaction, gasification and pyrolysis. Pyrolysis [4] is considered as a promising technology, which refers to the technology by which organic materials are decomposed into solid, liquid and gas products (bio-char, bio-oil, non-condensable gas) at high temperature (300–1000°C) in an inert atmosphere. Nowadays,

research on biomass pyrolysis mainly focuses on wood biomass, while little on aquatic biomass. Take algae as example, algae belong to low-grade, oxygen-releasing autotrophic plants, with varieties of species and wide distribution. Moreover, the output of many kinds of algae in China ranks first in the world. Most algae belong to single-celled organisms [5], which means it will be easy to be improved, and can be cultivated by changing the cultivation conditions to produce species more suitable for pyrolysis [6,7]. Compared with the first generation of biofuels [8] (edible biomass, sugar and starch plants) and the second generation of biomass [9] (lignocellulosic biomass), algae has several prominent advantages [10]: (i) high photosynthetic efficiency, which is conducive to alleviate the greenhouse effect problem; (ii) nutrients (N,P) can be extracted from wastewater and returned to the soil by waste product of fertilization; (iii) algae has a short breeding cycle [11], and its process of breeding is easy to realize automation; and (iv) algae does not need to occupy arable lands and is less affected by seasons and regions. According to the analysis composition of algae, algae has high lipid accumulation, which is suitable for the decomposition of bio-oil by heat conversion technology. Considering above advantages, algae is a potential biomass material.

There are always problems with biological oils prepared by direct pyrolysis of biological substances, such as low calorific value, high acid content and low hydrocarbon content. Therefore, measures should be taken to improve the quality of biological oils. One of the common methods is to use catalysts, such as molecular sieve, metals and metal oxides. Among them, metal oxide has large pore diameter, strong water stability and certain deoxidation performance, which is helpful to improve the stability of biological oil; alkali metals mainly include sodium salt, potassium salt, calcium salt and their oxides; microporous molecular sieve refers to molecular sieve with pore diameter less than 2 nm, which has good deoxidation and aromatization properties. HZSM-5 [12–14], one common molecular sieve, has a microporous structure that allows pyrolysis steam to enter the interior for further pyrolysis. Under the catalysis of HZSM-5, the release of oxygen-containing gas (CO and $CO_2$) shows a significant decline, and the conversion of furans to aromatic hydrocarbons may be promoted over strong acid site [12,13]. However, due to its small pore size [15], the yield of water and gas increased while the yield of organic matter decreased obviously. Besides, catalysis of HZSM-5 may be deactivated due to the polymerization of a mass of oxycompound [16]. The mesoporous zeolite MCM-41 with a larger pore size provides larger surface area and more accessible reaction sites. It has been pointed out that the mild acidity of MCM-41 catalyst provides an ideal environment for controlling the conversion of high molecular weight lignocellulosic molecules [13,15]. However, products will escape before complete re-pyrolysis because the pore size is too large [13]. Therefore, the utilization of fracture properties of macroporous catalysts and the re-framing properties of microporous molecular sieve catalysts as catalysts for biomass catalytic pyrolysis has received extensive attention [13]. In order to improve the quality of bio-oil from algae, existing studies have shown that oxygen content and acid compound of biological oil can be reduced when catalysed by Ni-supported zeolites [17], while nitrogen content of bio-oil can be reduced when catalysed by Mg–Al layered double oxide/ZSM-5 composites with an Mg/Al molar ratio of four (MgAl4 -LDO/ZSM-5) [18]. Hydrothermal carbonization [11] can increase the maximum weight loss rate of algae during pyrolysis and can be combined with catalytic pyrolysis to realize the reduction of nitrogen content in biological oil; ZSM-5 catalytic co-pyrolysis [19] can be a favourable process to enhance the yield of upgraded bio-oil. At present, composite molecular sieve catalyst is widely used in lignin biomass [20,21] and shows relatively superior catalytic performance. Previous studies have used the addition of macroporous and mesoporous molecular sieves to LOSA-1 to improve the selectivity of low-carbon olefins and aromatic hydrocarbons [13]. Meanwhile, the mixture of HZSM-5 and MCM-41 has been studied to improve the catalytic pyrolysis effect of fresh straw [22]. However, little research has been conducted on the composite of mesoporous and microporous molecular sieve and its application on proteinaceous biomass.

In this study, a hierarchical micro-mesoporous composite molecular sieve HZSM-5/MCM-41 with external mesoporous and internal microporous properties was developed through digestion and reassembly of molecular sieve HZSM-5, so as to meet re-pyrolysis requirements of pyrolysis steam in a wider range and improve the quality of pyrolysis products. With the support of pyrolysis-gas chromatography/mass spectrometry (Py-GC/MS) [23], this study intends to explore the catalytic pyrolysis products of algae and find suitable catalysts for catalytic pyrolysis to improve the quality of pyrolysis oil.

## 2. Material and methods

### 2.1. Experimental materials

The algae used in this study was spirulina, the selected molecular sieve catalyst was HZSM-5($SiO_2$/ $Al_2O_3$ ratio: 26 [24]), and the ratio of biomass to catalyst was 1 : 2 [13,23]. Pyrolysis products were

**Table 1.** The main characteristics of spirulina (ad, air-dry basis; A, ash; V, volatiles; FC, fixed carbon; M, moisture).

| | proximate analysis (wt%)[ad] | | | | ultimate analysis (wt%)[ad] | | | | |
|---|---|---|---|---|---|---|---|---|---|
| feedback | A | V | FC | M | C | H | O[a] | N | S |
| spirulina | 5.4 | 73.7 | 12.5 | 7.7 | 45.7 | 7.1 | 35.7 | 10.7 | 0.8 |

[a]Was calculated by difference.

determined by Py-GC/MS [23]. The main characteristics of spirulina can be seen in table 1. The content of oxygen was calculated from the difference of 100% and the mass ratio of C, H, N, S.

Tetra propylammonium hydroxide (TPAOH) was dissolved into 50 ml ultrapure water with a certain mass fraction to prepare TPAOH solution, and then 10 g HZSM-5 was added, stirred and heated in a water bath at 40°C for 1 h. Next cetyltrimethylammonium bromide (CTAB) solution of a certain concentration was mixed with HZSM-5 solution and heated in the water bath at 26°C for 1 h. Then, it was poured into the reactor for 24 h at a certain temperature for resolution. After adjusting the pH of the solution to 8.50 [2] by dilute sulfuric acid solution (5%), it was poured into the reactor for 24 h for crystallization. After the solution was filtered and dried, dried samples were heated at 550°C for 6 h, after which the preparation of catalyst composite molecular sieve HZSM-5/MCM-41 was completed.

## 2.2. Pyrolysis experiment

According to relevant studies, the temperature of pyrolysis by Pyroprobe 5200 pyrolyser [25] (CDS Analytical) in this study was 600°C [26,27] and the pyrolysis time was 20 s. First, 0.5 mg catalyst, 0.5 mg biomass raw material and 0.5 mg catalyst were poured into the quartz capillary tube successively, and quartz wool used to separate each layer and seal both ends. The heating rate was set at 20 000°C s$^{-1}$ [28], and the tube was kept at the reaction temperature for 20 s. High-purity helium (99.999%, Nanjing Maikesi Nanfen Special Gas Co., Ltd) was used as pyrolysis and carrier gas, with a constant flow rate of 1.0 ml min$^{-1}$. Products were identified by comparison with the NIST MS library database, and the analysis results were compared with the references to increase the reliability [20,28,29]. A DB-5 mas' capillary column (0.25 mm × 0.25 μm × 30 m) was used for product separation. The temperature in the oven was programmed to increase from 50°C (kept for 1 min) to 290°C (kept for 2 min) at a heating rate of 8°C min$^{-1}$. Pyrolysis of biomass raw materials, catalytic pyrolysis by HZSM-5 and composite molecular sieve HZSM-5/MCM-41 prepared under various preparation conditions were conducted, respectively. The analysis of organic composition was performed on a 7890A/5975C GC/MS analyser (Agilent Technologies).

## 2.3. Relative content

In the present work, the relative content ($R_{content}$) of organic pyrolytic products are calculated using a semi-quantitative method based on the area percentages of the chromatograph peaks and defined as follows [14,30]:

$$R_{content} = \frac{P_i}{P_{total}},$$

where $P_i$ is the peak area of a specific identified product, and $P_{total}$ is the total peak area under a certain condition.

# 3. Results and discussion

## 3.1. Influence of concentration of TPAOH on catalytic effect

In the preparation of catalyst, TPAOH solution is used to dissolve the crystal nucleus of HZSM-5, so that enough aluminosilicate is dissolved into the solution, preparing for the assembly of MCM-41 on the surface of HZSM-5 in the later stage. At the same time, digestion increases the specific surface area of HZSM-5, which is conductive to the formation of mesoporous structure on the surface.

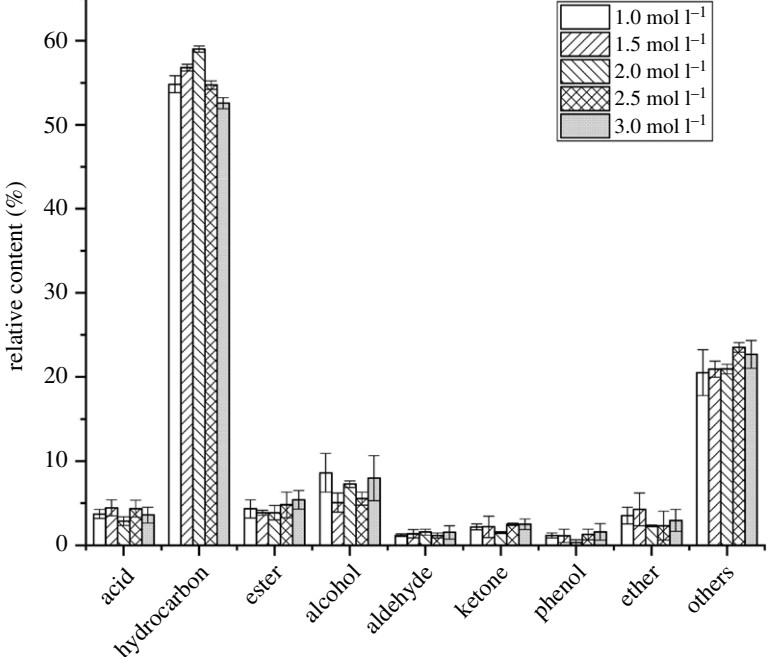

**Figure 1.** Distribution of pyrolysis products under catalysts with different TPAOH concentrations.

The concentration of TPAOH solution has an impact on the degree of digestion and affects the concentration of aluminosilicate in the solution and the specific surface area of HZSM-5 as a result.

In this study, concentrations of TPAOH solution in the preparation of composite molecular sieve HZSM-5/MCM-41s were selected as 1.0, 1.5, 2.0, 2.5 and 3.0 mol l$^{-1}$, respectively. As can be seen from figure 1, the change of the distribution of pyrolysis products follows the change of the concentration of TPAOH solution. As to the content of hydrocarbon, when the concentration of TPAOH is less than 2.0 mol l$^{-1}$, the content of hydrocarbon improves with the increase of the concentration and reaches a maximum of 59.17% at 2.0 mol l$^{-1}$. It may result from that the increase of concentration improves the concentration of aluminosilicate in the solution and the specific surface area of HZSM-5, which is conducive to the formation of composite molecular sieve HZSM-5/MCM-41. While the concentration of hydrocarbon products continues to increase, the content of hydrocarbon products declines. As to that of alcohol, when the concentration of TPAOH solution is 1.5 mol l$^{-1}$, 2.0 mol l$^{-1}$, 2.5 mol l$^{-1}$, the contents of alcohol are roughly the same of 7.5%. But when the concentration is 1.0 mol l$^{-1}$ or 3.0 mol l$^{-1}$, the content of alcohol can be found an apparent increase, reaching 13%. The reason for this phenomenon may be that the microporous structure of HZSM-5 begins to collapse with the increase of the concentration of TPAOH, and it is difficult to form an appropriate aperture for the complete re-pyrolysis of pyrolysis gas [31]. For other substances, the change of the concentration of TPAOH does not cause significant variation on their content.

## 3.2. Influence of the concentration of CTAB on catalytic effect

In the preparation process of composite molecular sieve HZSM-5/MCM-41, CTAB solution was used as template to establish the mesoporous structure on the surface of HZSM-5 during the crystallization process. The concentration of CTAB solution affects its interaction with aluminosilicate during the crystallization process and thereby influences the formation of MCM-41 outside HZSM-5.

The concentrations of CTAB solution chosen in the preparation of composite molecular sieve HZSM-5/MCM-41s in this study are 5, 10, 15, 20 wt%, respectively. As exhibited in figure 2, when the concentration of CTAB is below 10 wt%, the content of hydrocarbons in the pyrolysis product increases along with the increase of the concentration of CTAB in the solution. For this phenomenon, it may be because the increasing CTAB solution concentration makes for the formation of the orderly six-party structure and as a result forms a better structure of composite molecular sieve HZSM-5/MCM-41 after being demoulded. However, when the concentration of CTAB keeps growing, a decline can be seen in the content of hydrocarbons in the pyrolysis products. It is possible that the hexagonal

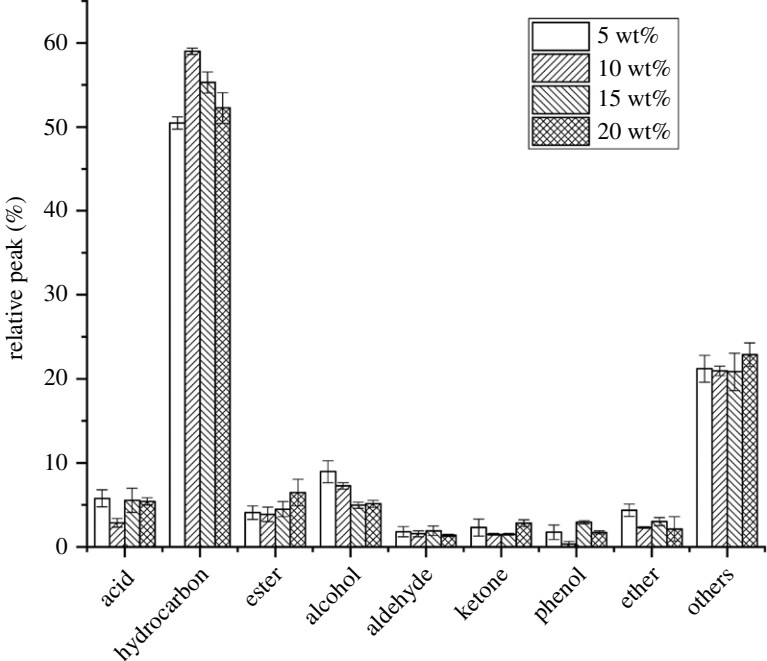

**Figure 2.** Distribution of pyrolysis products under catalysts with different CTAB concentrations.

structure starts to become disordered, and the damage to the microporous structure of HZSM-5 is excessive [32]. When the concentration of CTAB is 10 wt%, the damage to microporous structures of HZSM-5 is weaker, and the MCM-41 structure can be assembled on the surface. Therefore, its catalytic effect is the best, with the highest content of hydrocarbon of 59.17% and the lowest content of acid substances of 2.79%. At the same time, the content of aldehydes and ketones is low, which ensures the stability of pyrolysis products effectively.

## 3.3. Influence of different temperatures of crystallization on catalytic effect

The temperature of resolution during the preparation of the molecular sieve was 110°C, and the temperatures of crystallization were 90°C, 100°C, 110°C and 120°C.

As figure 3 shows, when the crystallization temperature is lower than 110°C, the increase of temperature promoted the increase of the content of hydrocarbons in the pyrolysis products. It may be because the increase of temperature promotes the reaction between dissolved crystal nucleus and template agent, which is conducive to the assembly of MCM-41 structures on the external surface of HZSM-5 and the formation of composite molecular sieve structures. However, as the temperature keeps increasing, decline can be seen in the content of hydrocarbon and increase can be found in the content of acids and alcohols. The explanation for this phenomenon may be that when the temperature further increases, the reaction between template ion and silicate root is unduly violent, which promotes the formation of amorphous material and destroys the assembly of mesoporous structures [30]. When the crystallization temperature is 110°C, the content of hydrocarbon reaches the highest (59.17%), while the contents of acids (2.79%), aldehydes (1.08%) and ketones (1.63%) reaches the lowest, which indicates that the deoxidation reaction is the most complete, and under this temperature of crystallization, the composite molecular sieve can have the most suitable structure for pyrolysis.

## 3.4. Influence of different digestion–crystallization time on catalytic effect

Digestion and crystallization are competing processes. The purpose of digestion is to digest HZSM-5 crystal nucleus under alkaline environment, and to improve its specific surface area to facilitate mesoporous structure assembly on its surface. At the same time, parts of the sialic acid radical dissolve into crystal fluid from crystal nucleus as a source of sialic acid radical to participate in the synthesis of MCM-41. The role of crystallization lies in the interaction between template molecule and silicon and aluminium source to form MCM-41.

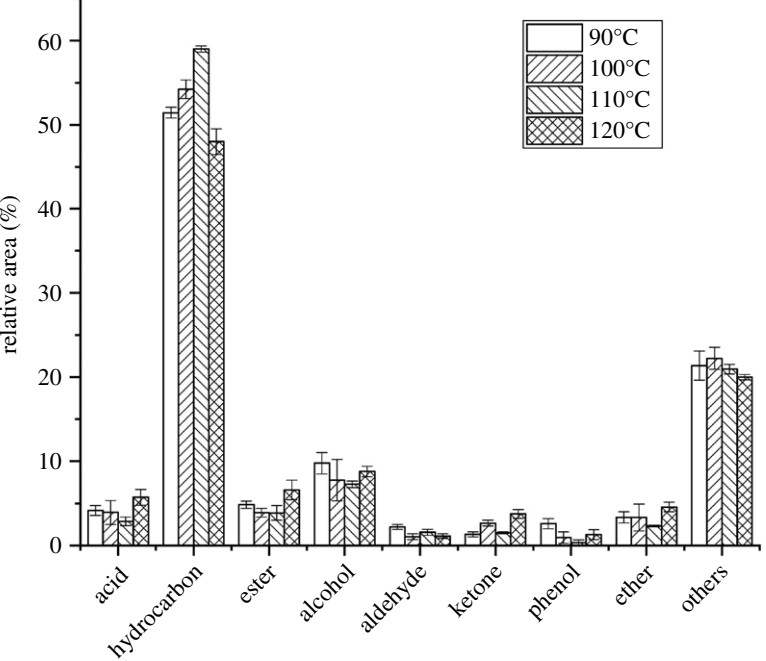

**Figure 3.** Distribution of catalyst products at different crystallization temperatures.

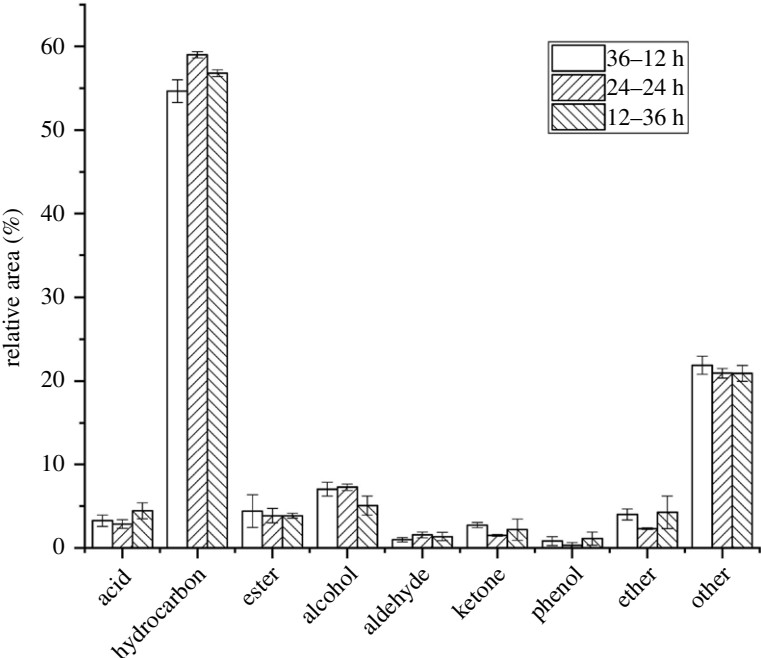

**Figure 4.** Distribution of catalyst products under different digestion–crystallization time.

In this study, the digestion–crystallization time was, respectively, adjusted to 12–36 h, 24–24 h, 36–12 h. As can be seen from figure 4, when the digestion time was 12 h and the crystallization time was 36 h, due to the short digestion time, the amount of dissolved sialic acid radical is small, which hindered the assembly of MCM-41 in the next step [30]. However, when the digestion time is 36 h and the crystallization time is 12 h, although a large amount of sialic acid radical is dissolved, the crystallization time is too short to fully assemble the mesoporous structure, and the order of the mesoporous structure is poor. When the digestion time and crystallization time are both 24 h, the mesoporous structure is formed and the microporous structure is less destroyed, and the ratio between the two competitive processes of digestion and crystallization is relatively appropriate. As can be seen from figure 4, when the digestion–crystallization time is 24–24 h, the hydrocarbon content reaches the highest level, and acids, aldehydes

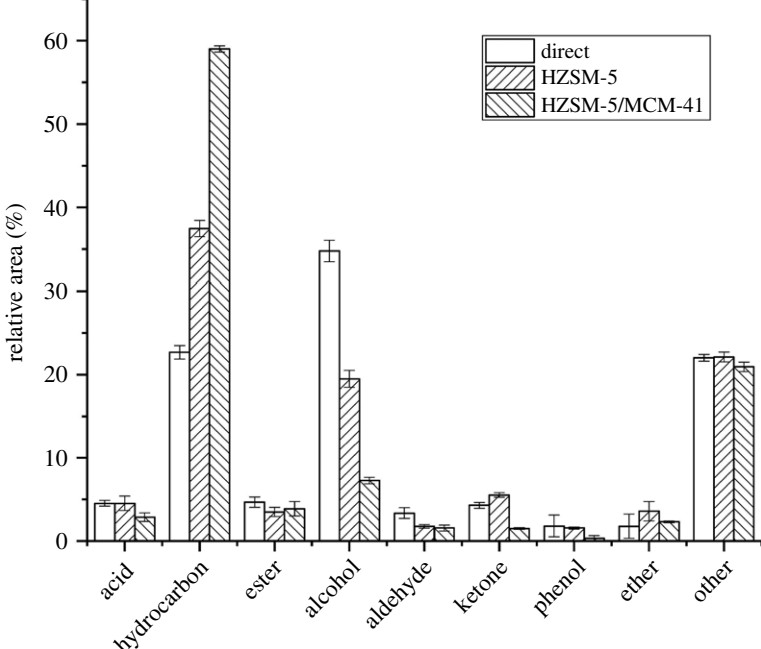

**Figure 5.** Distribution of pyrolysis products by direct pyrolysis, catalysed by HZSM-5 and composite molecular sieve HZSM-5/MCM-41.

and ketones all reach the lowest level, indicating that the composite molecular sieve HZSM-5/MCM-41 catalyst has a great catalytic pyrolysis effect.

## 3.5. Comparison of pyrolysis and catalytic pyrolysis products

In this study, direct pyrolysis of algae and catalytic pyrolysis by HZSM-5 and composite molecular sieve HZSM-5/MCM-41 were conducted. The distribution of pyrolysis products is shown in figure 5.

In terms of direct pyrolysis, due to the high content of nitrogen in algae, the content of nitrogen-containing organic compounds is significantly higher than that of herbaceous biomass, which is consistent with the previous research [26]. Due to the low lignin content of algae, the phenolic substances in their pyrolysis products are also few, accounting for only 3%. Hydrocarbon content is relatively high, up to 25%, and so as to alcohols, up to 30%, which may be due to the high axunge content of algae themselves. At the same time, contents of acids and aldehydes were also lower, at 5% and 5%, respectively. It can be found that the bio-oil generated by direct pyrolysis of algae has high quality, indicating that algae is a promising biomass raw material.

In this study, preparation conditions of HZSM-5/MCM-41 composite molecular sieve were determined as: TPAOH solution: $2.0$ mol $l^{-1}$, CTAB solution: 10 wt%, crystallization temperature: 110°C, digestion–crystallization time: 24–24 h. Compared with the catalytic pyrolysis by HZSM-5, for the micropore in HZSM-5, pyrolysis products were deoxidized in re-pyrolysis, the hydrocarbon content in the products is significantly increased, alcohols are significantly reduced, and aldehydes are also reduced to some extent. Catalysed by HZSM-5, acid substances increase slightly, from 4 to 5%; the hydrocarbon content increases significantly from 15 to 28%; alcohols decrease significantly from 35 to 20%; the content of aldehydes also drops, from 5 to 2%; esters also fall, from 5 to 3%. It can be considered that through the catalytic pyrolysis of HZSM-5, due to the appropriate pore size of HZSM-5, parts of oxygen-containing organic compounds further undergo deoxidation during re-pyrolysis, and main components are some alcohols of macromolecules [27,33]. The content of nitrogen-containing compounds is hardly affected, indicating that HZSM-5 could not promote the removal of nitrogen from nitrogen-containing organic compounds.

Compared with HZSM-5, when catalysed by composite molecular sieve HZSM-5/MCM-41, the content of acid substances decreased from 5 to 2.17%; the hydrocarbon content increased significantly from 38 to 59.17%; alcohols were further reduced from 20 to 8%; aldehyde and ketone contents also declined, from 3 and 5% to 2 and 2%, respectively. It can be considered that due to the special structure of the external mesoporous and internal micropores of the composite molecular sieve HZSM-5/MCM-41, more oxygen-containing organic substances undergo deoxidation reaction during further pyrolysis inside the catalyst, while alcohols are also the main organic constituents for re-pyrolysis. Similarly, the content of

**Table 2.** Content of alcohols and hydrocarbons in the products of direct pyrolysis, catalytic pyrolysis by HZSM-5 and HZSM-5/ MCM-41 of algae.

| compound | direct (%) | HZSM-5 (%) | HZSM-5/MCM-41 (%) |
|---|---|---|---|
| hydrocarbons | | | |
| pentane | 2.15 | 0.00 | 0.00 |
| cyclopropane, 1,2-dimethyl-, cis- | 0.00 | 5.55 | 9.30 |
| 2-hexene, (Z)- | 0.00 | 0.00 | 2.92 |
| 2,4-hexadiene, (Z, Z)- | 0.00 | 0.00 | 1.21 |
| 3-vinyl-1-cyclobutene | 0.66 | 0.00 | 0.00 |
| benzene | 0.00 | 1.27 | 2.95 |
| 3-hexene, 2-methyl-, (Z)- | 0.00 | 0.39 | 0.00 |
| 2-heptene | 0.00 | 0.47 | 0.00 |
| 2-hexene, 5-methyl-, (E)- | 0.83 | 0.52 | 0.00 |
| 1,4-hexadiene, 2-methyl- | 0.00 | 0.00 | 1.62 |
| toluene | 5.82 | 5.70 | 5.47 |
| 3-undecene, (Z)- | 0.00 | 0.76 | 0.00 |
| ethylbenzene | 1.06 | 1.15 | 1.35 |
| p-xylene | 0.00 | 3.12 | 4.52 |
| styrene | 0.00 | 0.00 | 0.90 |
| 1,3,5,7-cyclooctatetraene | 0.87 | 0.90 | 0.00 |
| benzene, propyl- | 0.00 | 0.00 | 0.47 |
| benzene, 1-ethyl-3-methyl- | 0.00 | 0.88 | 0.00 |
| benzene, 1-ethyl-4-methyl- | 0.00 | 0.00 | 2.08 |
| benzene, 1,3,5-trimethyl- | 0.00 | 0.00 | 0.40 |
| D-limonene | 0.69 | 0.23 | 0.23 |
| 1,3,5-cycloheptatriene, 7-ethyl- | 0.00 | 0.05 | 0.00 |
| benzene, 1-ethenyl-2-methyl- | 0.00 | 0.00 | 0.44 |
| indene | 0.00 | 0.00 | 0.60 |
| benzene, 1,2-diethyl- | 0.00 | 0.00 | 1.56 |
| 1-phenyl-1-butene | 0.00 | 0.00 | 0.33 |
| 3a,6-methano-3aH-indene, 2,3,6,7-tetrahydro- | 0.00 | 0.00 | 0.48 |
| 1H-indene, 3-methyl- | 0.00 | 0.00 | 0.38 |
| naphthalene | 0.00 | 0.00 | 0.95 |
| bicyclo[4.4.1]undeca-1,3,5,7,9-pentaene | 0.00 | 0.37 | 0.78 |
| bicyclopentyl-1,1'-diene | 0.00 | 1.05 | 0.00 |
| tricyclo[6.4.0.0(3,7)]dodeca-1,9,11-triene | 0.00 | 0.00 | 0.68 |
| alcohols | | | |
| cyclopropaneethanol | 6.81 | 0.00 | 0.00 |
| 2-octyn-1-ol | 0.00 | 0.00 | 0.35 |
| trans-2-ethyl-2-hexen-1-ol | 0.00 | 0.00 | 0.33 |
| 3-octyn-1-ol | 0.00 | 1.97 | 0.00 |
| 4-penten-1-ol, 2-methylene- | 0.00 | 0.88 | 0.00 |
| 3-pentyn-1-ol | 1.04 | 0.40 | 0.00 |

**Table 2.** (*Continued.*)

| compound | direct (%) | HZSM-5 (%) | HZSM-5/MCM-41 (%) |
|---|---|---|---|
| 3-decyn-2-ol | 0.00 | 0.00 | 0.34 |
| 1-cyclohexene-1-methanol | 0.00 | 0.72 | 0.00 |
| 2,5-cyclooctadien-1-ol | 0.00 | 0.00 | 0.37 |
| 2-decyn-1-ol | 0.00 | 0.06 | 0.17 |
| 2-undecanethiol, 2-methyl- | 0.00 | 0.13 | 0.00 |
| 4-cyclopentene-1,3-diol, trans- | 1.63 | 0.00 | 0.00 |
| 4-penten-1-ol, 2-methylene- | 1.16 | 0.00 | 0.00 |
| 5-hexen-2-ol | 0.42 | 0.00 | 0.00 |
| 3-buten-2-ol, 3-methyl- | 0.90 | 0.00 | 0.00 |
| 3-hexyne-2,5-diol | 0.90 | 0.00 | 0.00 |
| 2-nonen-1-ol, (E)- | 0.26 | 0.00 | 0.00 |
| bicyclo[2.1.1]hexan-2-ol, 2-ethenyl- | 0.61 | 0.00 | 0.00 |
| cyclohexanol, 2-methyl-5-(1-methylethenyl)- | 0.08 | 0.00 | 0.00 |
| isopinocarveol | 0.27 | 0.38 | 0.00 |
| 2-nitrohept-2-en-1-ol | 0.19 | 0.00 | 0.00 |
| cis-*p*-mentha-2,8-dien-1-ol | 0.85 | 0.00 | 0.00 |
| 1-(4-methoxyphenyl)-2-(4-pyridazinyl) ethanol | 0.00 | 0.43 | 0.00 |
| 2-naphthol, 1,2,3,4,4a,5,6,7-octahydro-4a-methyl- | 0.59 | 0.37 | 0.00 |
| 5,7-dodecadiyn-1,12-diol | 0.23 | 0.29 | 0.40 |
| 1,2,4-metheno-1H-cyclobuta[cd]pentalene-3,5-diol, octahydro- | 0.00 | 0.00 | 0.45 |
| (1-benzyl-cyclopropyl)-methanol | 0.00 | 0.00 | 0.97 |
| falcarinol | 0.28 | 0.48 | 0.51 |
| ethanol, 2-(9-octadecenyloxy)-, (Z)- | 3.29 | 3.19 | 1.02 |
| estra-1,3,5(10)-trien-17πol | 0.00 | 1.20 | 0.28 |

nitrogenous organic compounds is not affected, indicating that the removal of nitrogen could not be achieved by just adjusting the aperture.

The contents of alcohols and hydrocarbons in direct pyrolysis and catalytic pyrolysis products are shown in table 2 [34]. Compared with direct pyrolysis products, micromolecular alcohol content in pyrolysis products catalysed by HZSM-5 shows a significant decline, for example, the content of cyclopropane ethanol, 3-pentyn-1-ol and 4-cyclopentene-1,3-diol, trans- decreases respectively from 6.8, 1.04 and 1.63% to 0, 0.4 and 0%, which indicates that sufficient hydrogenation, deoxidization, dehydration reaction happens to these compounds. The addition of HZSM-5 helps to reduce the energy barrier due to hydrogen bond and van der Waals force, thus promoting the occurrence of repyrolysis. And the possible pathways [35,36] of their hydrodeoxidation reactions are shown in figure 6. As to macromolecular alcohols such as ethanol, 2-(9-octadecenyloxy)-, (Z)-, because of the limitation of the aperture of microporous structure, HZSM-5 failed to show great catalytic pyrolysis performance. HZSM-5/MCM-41 shows great catalytic properties on micromolecular alcohols due to its internal microporous structure and strong acid sites. It also has hydrodeoxidation effect on macromolecular alcohols for its external mesoporous structure. For hydrocarbon products, compared with HZSM-5, the content of each product increased under the catalysis of HZSM-5/MCM-41, which could be attributed to the fact that the crust-core structure of HZSM-5/MCM-41 increased the residence time of pyrolysis gas in the structure and thus realized more sufficient hydrodeoxidation performance. Compared with bio-char, Ni/SBA-15 [18] and activated carbon [11], HZSM-5/MCM-41 demonstrated excellent deoxidation performance on alcohols; compared with composite molecular sieve prepared with NaOH [29], the reaction of digestion using TPAOH is mild and easy to be

**Figure 6.** The possible hydrodeoxidation pathways of partial alcohols.

controlled. Previous studies involving catalysts modified by an alkali treatment have focused on alkali-treated HZSM-5 at low concentration (less than 1.0 mol l$^{-1}$) [29] of inorganic base. The high concentration (2.0 mol l$^{-1}$) of TPAOH solution is more effective for HZSM-5 treatment than the low concentration base solution (less than 1.0 mol l$^{-1}$). Moreover, the modified catalyst does not require NH$_4$NO$_3$-ion exchange. The 2.0 mol l$^{-1}$ TPAOH modified HZSM-5 led to produce the highest aromatic selectivity and catalytic activity from CFP of algae.

## 3.6. Microstructure of HZSM-5 and HZSM-5/MCM-41

In order to study the effect of digestion and crystallization on the microstructure of HZSM-5, SEM characterization of catalyst samples was performed. Figure 7a is the SEM figure of molecular sieve HZSM-5. It can be seen from the figure that the HZSM-5 is of regular cubic shape with smooth

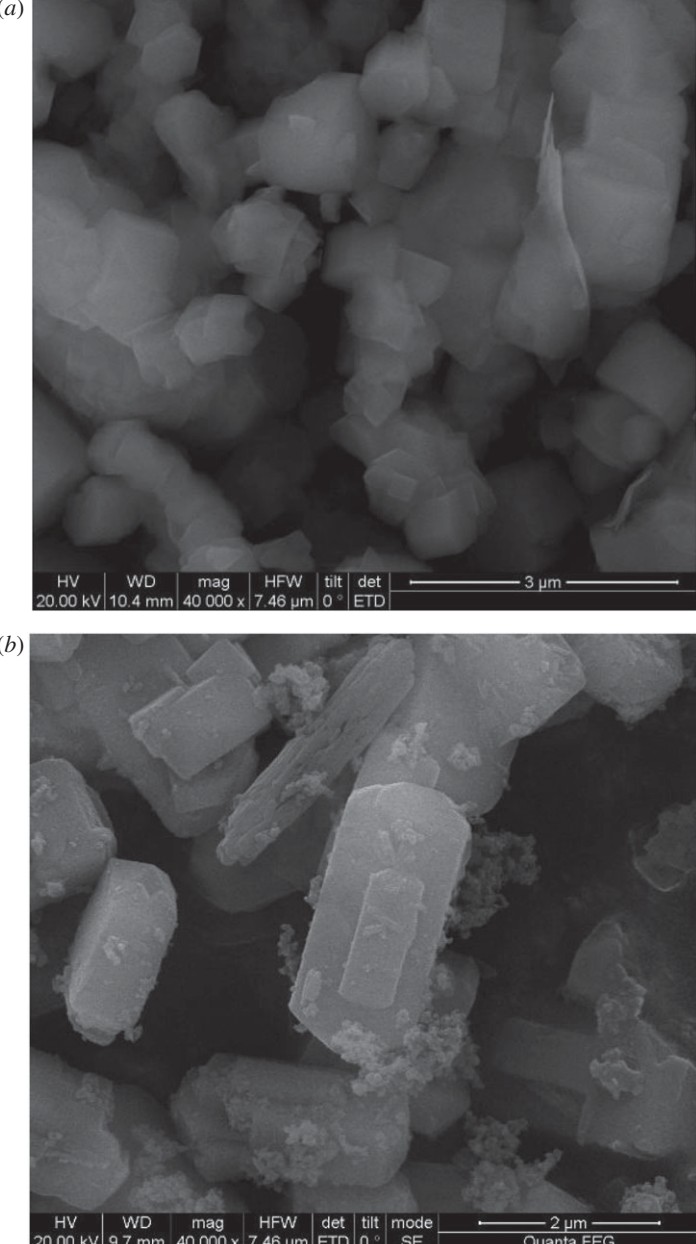

**Figure 7.** (*a*) SEM of HZSM-5. (*b*) SEM of HZSM-5/MCM-41.

surface. Figure 7*b* shows the SEM figure of HZSM-5/MCM-41 composite molecular sieve HZSM-5/MCM-41. It can be seen that the surface becomes fragmented and irregular, and many small particles are generated to adhere to the surface of crystal nucleus, namely, the synthetic MCM-41. As can be seen from the figure, compared with the molecular sieve of HZSM-5/MCM-41, the surface pore diameter of the composite molecular sieve HZSM-5/MCM-41 of HZSM-5 is significantly increased, and at the same time, it can maintain a regular overall structure without excessive collapse of the skeleton, which may be the reason for its better catalytic deoxidation effect.

## 4. Conclusion

By investigating the preparation conditions of HZSM-5/MCM-41, it is determined that under the condition of TPAOH concentration $2.0 \, \text{mol} \, l^{-1}$, CTAB concentration 10 wt%, crystallization temperature 110°C, digestion–crystallization time 24–24 h, the external mesoporous structure of the composite molecular sieve can be sufficiently formed and the internal microporous structure would

not be excessively damaged. The main components in the direct pyrolysis products of algae are alcohols, followed by N-containing compounds and hydrocarbons, while the adverse components aldehydes, ketones and acids have low contents, which indicate the bio-oils are of high quality. Compared with direct pyrolysis, catalytic pyrolysis by HZSM-5 can effectively reduce the oxygen content of pyrolysis products, mainly by reducing the content of alcohols. However, the adverse components do not change significantly due to their low content, and the content of N-containing compounds is hardly affected. After catalytic pyrolysis by HZSM-5/MCM-41, the alcohols decrease significantly again, and the content of adverse components also decrease, but the content of nitrogen-containing organic matter is not affected. After analysing the specific substances, it is found that HZSM-5 had great hydrodeoxidation performance on micromolecular alcohols, but the catalytic effect was limited by its pore size. The external mesoporous structure of HZSM-5/MCM-41 not only widens the reaction range of hydrodeoxidation reaction, but also increases the residence time of pyrolysis products in the internal structure, making deoxidation hydrogenation reaction more complete. Through the analysis of the products, it is found that the alcohols are mainly converted into hydrocarbons by hydrogenation, deoxidation and dehydration under the catalysis. Finally, high nitrogen content is the problem of algae pyrolytic oil. The composite molecular sieve has no effect on nitrogen content although its deoxidation effect is obvious. How to reduce the nitrogen content in the pyrolysis products of algae and its combination with composite molecular sieves to improve the quality of algae pyrolysis products can be one of the later research directions.

Data accessibility. Electronic supplementary material data are available within the Dryad Digital Repository: https://doi.org/10.5061/dryad.22829q7 [37].
Authors' contributions. H.Z. analysed the results and drafted the manuscript. Z.L. collected the samples, recorded the spectra and processed the spectra tighter with H.Z. W.W. participated in all the tests. Z.L. helped with interpretation of the data. Z.Z. revised the manuscript. W.W. reviewed the draft.
Competing interests. We declare we have no competing interests.
Funding. This work was sponsored by the National Natural Science Foundation of China (grant no. 51776042).

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
