## [Reviewer comments · Royal Society Open Science]

Review History

RSOS-191307.R0 (Original submission)

Review form: Reviewer 1

Is the manuscript scientifically sound in its present form?

Yes

Are the interpretations and conclusions justified by the results?

Yes

Is the language acceptable?

Yes

Do you have any ethical concerns with this paper?

No

Have you any concerns about statistical analyses in this paper?

Yes

Recommendation?

Accept with minor revision (please list in comments)

Comments to the Author(s)

This manuscript describes that effect of hierarchical micro-mesoporous composite molecular sieve HZSM-5 / MCM-41 zeolite on the quality of bio-oil from Py-GC/MS of algae. After reading the manuscript, I think it is a very interesting work that deserves publication in this journal. It is should be accepted after reasonably reply and revise the following suggestions:

1. In the analysis section, additional references are needed to further explain the changes in catalytic performance;
2. Please repeat the experiment at least three times and add an error bar to the diagram;
3. In conclusion section, please add the comparison with other references to illustrate the superiority of catalysts in this study;
4. Conduct elemental and industrial analysis on algae;
5. Why not use quantitative analysis and only use peak area to explain the results;
6. The database can not only use NIST MS library database, but also need to be compared with other references ;
7. Please amend the language in the article.

Review form: Reviewer 2 (Abdulkareem Ghassan Alsultan)

Is the manuscript scientifically sound in its present form?

No

Are the interpretations and conclusions justified by the results?

No

Is the language acceptable?

Yes

Do you have any ethical concerns with this paper?

No

Have you any concerns about statistical analyses in this paper?

No

Recommendation?

Major revision is needed (please make suggestions in comments)

Comments to the Author(s)

- 1- What were the highlight of your work?
- 2- What is your research novelty?
- 3- The introduction part need to rewrite using updated references and the most important articles on your research area
- 4- Summary need to be rewritten in more scientific way.
- 5- What is the heating rate is it 20000 or 20C/s
- 6- Are the reported results (not including the characterization results) in your paper reproducible? Are those come from a single experiment since the number of runs are not clearly indicated.

7- The discussion seems to be superficial, therefore more in-deep discussion is required to improve the scientific aspect .

Conclusions. Some critical guidance in the evaluation of the present state and in the further study is given in this section.

Decision letter (RSOS-191307.R0)

16-Sep-2019

Dear Dr Zhao:

Title: Research on catalytic pyrolysis of algae based on Py-GC/MS

Manuscript ID: RSOS-191307

The editor assigned to your manuscript has now received comments from reviewers. We would like you to revise your paper in accordance with the referee and Subject Editor suggestions which can be found below (not including confidential reports to the Editor). Please note this decision does not guarantee eventual acceptance.

Please submit your revised paper before 09-Oct-2019. Please note that the revision deadline will expire at 00.00am on this date. If we do not hear from you within this time then it will be assumed that the paper has been withdrawn. In exceptional circumstances, extensions may be possible if agreed with the Editorial Office in advance. We do not allow multiple rounds of revision so we urge you to make every effort to fully address all of the comments at this stage. If deemed necessary by the Editors, your manuscript will be sent back to one or more of the original reviewers for assessment. If the original reviewers are not available we may invite new reviewers.

Please also include the following statements alongside the other end statements. As we cannot publish your manuscript without these end statements included, if you feel that a given heading is not relevant to your paper, please nevertheless include the heading and explicitly state that it is not relevant to your work.

- Acknowledgements

RSC Associate Editor:
Comments to the Author:
(There are no comments.)

RSC Scientific Editor:
Comments to the Author:
(There are no comments.)

Reviewers' Comments to Author:
Reviewer: 1

Comments to the Author(s)

This manuscript describes that effect of hierarchical micro-mesoporous composite molecular sieve HZSM-5 / MCM-41 zeolite on the quality of bio-oil from Py-GC/MS of algae. After reading the manuscript, I think it is a very interesting work that deserves publication in this journal. It is should be accepted after reasonably reply and revise the following suggestions:

1. In the analysis section, additional references are needed to further explain the changes in catalytic performance;
2. Please repeat the experiment at least three times and add an error bar to the diagram;
3. In conclusion section, please add the comparison with other references to illustrate the superiority of catalysts in this study;
4. Conduct elemental and industrial analysis on algae;
5. Why not use quantitative analysis and only use peak area to explain the results;
6. The database can not only use NIST MS library database, but also need to be compared with other references ;
7. Please amend the language in the article.

Reviewer: 2

Comments to the Author(s)

1- What were the highlight of your work?

2- What is your research novelty?

3- The introduction part need to rewrite using updated references and the most important articles on your research area

4- Summary need to be rewritten in more scientific way.

5- What is the heating rate is it 20000 or 20C/s

6- Are the reported results (not including the characterization results) in your paper reproducible? Are those come from a single experiment since the number of runs are not clearly indicated.

7- The discussion seems to be superficial, therefore more in-deep discussion is required to improve the scientific aspect .

Conclusions. Some critical guidance in the evaluation of the present state and in the further study is given in this section.

Author's Response to Decision Letter for (RSOS-191307.R0)

See Appendices A - C.

Decision letter (RSOS-191307.R1)

15-Oct-2019

Dear Dr Zhao:

Title: Research on catalytic pyrolysis of algae based on Py-GC/MS

Manuscript ID: RSOS-191307.R1

It is a pleasure to accept your manuscript in its current form for publication in Royal Society Open Science. The chemistry content of Royal Society Open Science is published in collaboration with the Royal Society of Chemistry.

Royal Society of Chemistry
Thomas Graham House

Science Park, Milton Road
Cambridge, CB4 0WF
Royal Society Open Science - Chemistry Editorial Office

RSC Associate Editor
Comments to the Author:
(There are no comments.)

Reviewer(s)' Comments to Author:

Appendix A

Reviewers/Editor comments:

Reviewer #1: This manuscript describes that effect of hierarchical micro-mesoporous composite molecular sieve HZSM-5 / MCM-41 zeolite on the quality of bio-oil from Py-GC/MS of algae. After reading the manuscript, I think it is a very interesting work that deserves publication in this journal. It should be accepted after reasonably reply and revise the following suggestions:

1. In the analysis section, additional references are needed to further explain the changes in catalytic performance;

Thank you very much for your kind comment.

Compared with direct pyrolysis, the most important changes of catalytic pyrolysis products are the decrease of alcohols and the increase of hydrocarbons. Therefore, some literatures on the mechanism and reaction path of catalytic pyrolysis of alcohols to produce hydrocarbons were referred to, and the conversion mechanism and path of substances in this study were speculated based on these studies; The influence of TPAOH concentration change on catalyst structure in the early stage was not well explained. Therefore, some literatures on catalytic performance of catalysts after TPAOH treatment were referred to to illustrate the influence of TPAOH concentration change on catalyst structure.

References:

M. MacLean, J. Nadeau, T. Gurnea, C. Tittiger and G. J. Blomquist, *Insect Biochem Mol Biol*, 2018, 102, 11-20. (doi: 10.1016/j.ibmb.2018.09.005)

D. Jiang, Z. Xia, S. Wang, H. Li, X. Gong, C. Yuan, A. El-Fatah Abomohra, B. Cao, X. Hu, Z. He and Q. Wang, *Journal of Analytical and Applied Pyrolysis*, 2019, 143. (doi: 10.1016/j.jaap.2019.104680)

Z. Li, E. Jiang, X. Xu, Y. Sun and R. Tu, *Renewable Energy*, 2020, 146, 1991-2007. (doi: 10.1016/j.renene.2019.08.012)

Y. Jiao, X. Fan, M. Perdjon, Z. Yang and J. Zhang, *Applied Catalysis A: General*, 2017, 545, 104-112. (doi: 10.1016/j.apcata.2017.07.036)

S. Tang, C. Zhang, X. Xue, Z. Pan, D. Wang and R. Zhang, *Journal of Analytical and Applied Pyrolysis*, 2019, 137, 86-95. (doi: 10.1016/j.jaap.2018.11.013)

Y. Qiao, B. Wang, P. Zong, Y. Tian, F. Xu, D. Li, F. Li and Y. Tian, *Energy Conversion and Management*, 2019, 199. (doi: 10.1016/j.enconman.2019.111964)

2. Please repeat the experiment at least three times and add an error bar to the diagram;

Answer:

Thank you very much for your kind comment.

In order to demonstrate the repeatability of the experimental results, two repeated experiments were carried out, and the experimental results were consistent with the first experiment. We integrated the results of three experiments, took the average value and added error bars to each group of data to explain the experimental results, so as to increase the credibility of the experimental results.

3. In conclusion section, please add the comparison with other references to illustrate the superiority of catalysts in this study;

Thank you very much for your kind comment.

The superiority of the catalyst has been partly explained in the introduction part. In the revised draft, the comparison of the catalytic performance between the catalyst in this research those reported in the literature in recent years has been added in the discussion part. While maintaining a good catalytic effect, HZSM-5 / MCM-41 expanded the scope of pyrolysis and reduced the occurrence of surface coking. From the experimental results, it can be found that the composite molecular sieve catalyst has not only good catalytic effect on micromolecular alcohol, but also on macromolecular alcohol, showing the superiority of the composite molecular sieve.

Compared with metal catalyst in other researches, HZSM-5/MCM-41 is of lower cost; Compared with bio-char, Ni/SBA-15 and activated carbon, HZSM-5/MCM-41 demonstrated excellent deoxidation performance on alcohols; Compared with composite molecular sieve prepared with NaOH, the reaction of digestion using TPAOH is mild and easy to be controlled.

References:

X. Zhang, C. Li, A. Tian, Q. Guo and K. Huang, Carbon Resources Conversion, 2019, 2, 111-116. (doi: 10.1016/j.crcon.2019.05.002)

H.Zhang, R. Xiao, B. Jin, G. Xiao and R. Chen, Bioresour Technol, 2013, 140, 256-262. (doi: 10.1016/j.biortech.2013.04.094)

D.Wang, R. Xiao, H. Zhang and G. He, Journal of Analytical and Applied Pyrolysis, 2010, 89, 171-117. (doi: 10.1016/j.jaap.2019.04.014)

L. Gao, J. Sun, W. Xu and G. Xiao, Bioresour Technol, 2017, 225, 293-298. (doi: 10.1016/j.biortech.2016.11.077)

E.Ansah, L. Wang, B. Zhang and A. Shahbazi, Fuel, 2018, 228, 234-242. (doi: 10.1016/j.fuel.2018.04.163)

S.Tang, C. Zhang, X. Xue, Z. Pan, D. Wang and R. Zhang, Journal of Analytical and Applied Pyrolysis, 2019, 137, 86-95. (doi: 10.1016/j.jaap.2018.11.013)

4. Conduct elemental and industrial analysis on algae;

Elemental and industrial analysis has been conducted on our material, and the result has been showed in our article. Among them, the content of O was calculated from the difference of 100% and the mass ratio of C, H, N, S.

5. Why not use quantitative analysis and only use peak area to explain the results;

The integration parameters were set as follows: the initial peak width is 0.110, initial area reject as 0, and shoulder detection off. In accordance to procedure previously established in our lab, we varied the initial threshold with the goal of limiting the number of detected compounds to around 80 (± 3). The values were: 16.0 for direct pyrolysis, 17 for catalytic pyrolysis by HZSM-5, and 16.5 for catalytic pyrolysis by

HZSM-5/MCM-41 prepared under various conditions, and 19.2 for the catalytic experiments. The large amount of compounds quantified (80) ensures that the majority of the products are accounted for. Any peaks that are not among the top 105 in area size are too small to affect the results in any significant way and were therefore neglected.

The quantification method in the present paper assumes the compounds identified by GC-MS to have a similar response factors, resulting in the use of total areas as proxies for yields. This assumption is justified by the similar nature of the products obtained in each case, since the products of non-catalytic pyrolysis are typically oxygenated organic molecules, and the products of catalytic pyrolysis are typically hydrocarbons. Therefore, we expect that differences in calibration response factors for compounds in a single experiment are not large enough to cause significant deviations in the results, in such a way that the trends and conclusions from the experiments reported in the present work are representative of the process.

6. The database can not only use NIST MS library database, but also need to be compared with other references;

Thank you very much for your kind comment.

In order to improve the credibility of the database, we refer to GC/MS analysis data in some published literatures on the basis of referring to the NIST MS library database. By comparing the GC/MS data with the compounds corresponding to the peaks in the GC/MS data at various time points in the literature, we can determine the accuracy of the database and the credibility of the experimental results.

References:

Zou, Y. Wang, L. Jiang, Z. Yu, Y. Zhao, Q. Wu, L. Dai, L. Ke, Y. Liu and R. Ruan, Bioresour Technol, 2019, 289, 121609. (doi:10.1016/j.biortech.2019.121609)

Tang, C. Zhang, X. Xue, Z. Pan, D. Wang and R. Zhang, Journal of Analytical and Applied Pyrolysis, 2019, 137, 86-95. (doi: 10.1016/j.jaap.2018.11.013)

K. Ding, Z. Zhong, J. Wang, B. Zhang, L. Fan, S. Liu, Y. Wang, Y. Liu, D. Zhong, P. Chen and R. Ruan, Bioresour Technol, 2018, 261, 86-92.(doi: 10.1016/j.biortech.2018.03.138)

7. Please amend the language in the article.

The conclusion part of the original manuscript lacks scientific nature, and the analysis part of the original manuscript has been modified to add the part of mechanism analysis, so we have rewritten the conclusion part according to the current analysis and discussion part. In addition, we have corrected other unqualified parts of languages.

Appendix B

Reviewer: 2 Comments to the Author(s)

1- What were the highlight of your work?

The most important part of this study is the application of composite molecular sieve catalyst in algae pyrolysis. Algae is considered as a new generation of biomass fuel with potential, and the content of oxygen compounds in its direct pyrolysis products is high, which is not conducive to the direct utilization of bio-oil. In order to reduce the oxygen content of its pyrolysis products, we applied the composite molecular sieve to the catalytic pyrolysis of algae, and evaluated the performance of the catalyst according to the analysis results of the pyrolysis products of algae, so as to determine the best conditions of preparation of the catalyst.

At the same time, through the analysis of the products, it is found that catalytic pyrolysis is mainly to deoxidize and hydrogenate the alcohols in the products. Through the catalytic pyrolysis of the composite molecular sieve catalyst, the content of hydrocarbons in the catalytic products increased from 20% to nearly 60%, while the content of the adverse components remained at a very low level, indicating that the bio-oil had a high quality.

2- What is your research novelty?

At present, there are many studies on the catalytic pyrolysis of composite molecular sieve catalysts in lignin biomass, but its application in algae pyrolysis is not enough. However, lignin biomass is quite different from algal biomass in composition, and the results of catalytic pyrolysis of many lignin biomasses may not be applicable to algal biomass. In preparation of catalyst, different from NaOH used in previous studies, this study used mild alkaline TPAOH to modify HZSM-5. Previous studies involving catalysts modified by an alkali treatment have focused on alkali-treated HZSM-5 at low concentration (<1.0 mol/L) of inorganic base. The high concentration (2.0 mol/L) of TPAOH solution is more effective for HZSM-5 treatment than the low concentration base solution (< 1.0 mol/L). Moreover, the modified catalyst does not require NH₄NO₃ ion exchange. The 2.0 mol/L TPAOH modified HZSM-5 led to produce the highest aromatic selectivity and catalytic activity from CFP of algae. In addition, the higher concentration such as 2.0mol/L was used to improve the reaction rate, so as to ensure the digestion effect and improve the preparation efficiency.

3- The introduction part need to rewrite using updated references and the most important articles on your research area

Thank you very much for your kind comment.

We refer to the researches on the catalytic pyrolysis of algae and the catalysis of composite molecular sieve in the past three years, especially the influential reports on such researches in some authoritative journals and learn the current research achievements in this field. At the same time, the statement of the premature research results in the introduction is deleted and the research achievements in recent years are sorted out and supplemented in the introduction section to improve the timeliness and credibility of this article.

References:

N. H. Zainan, S. C. Srivatsa, F. Li and S. Bhattacharya, Fuel, 2018, 223, 12-19. (doi: 10.1016/j.fuel.2018.02.166)

L. Gao, J. Sun, W. Xu and G. Xiao, Bioresour Technol, 2017, 225, 293-298. (doi: 10.1016/j.biortech.2016.11.077)

E. Ansah, L. Wang, B. Zhang and A. Shahbazi, Fuel, 2018, 228, 234-242. (doi: 10.1016/j.fuel.2018.04.163)

Y.-M. Kim, J. Jeong, S. Ryu, H. W. Lee, J. S. Jung, M. Z. Siddiqui, S.-C. Jung, J.-K. Jeon, J. Jae and Y.-K. Park, Energy Conversion and Management, 2019, 195, 727-737. (doi: 10.1016/j.enconman.2019.05.034)

R. Zou, Y. Wang, L. Jiang, Z. Yu, Y. Zhao, Q. Wu, L. Dai, L. Ke, Y. Liu and R. Ruan, Bioresour Technol, 2019, 289, 121609. (doi: 10.1016/j.biortech.2019.121609)

B. Cao, Z. Xia, S. Wang, A. E.-F. Abomohra, N. Cai, Y. Hu, C. Yuan, L. Qian, L. Liu, X. Liu, B. Li, Z. He and Q. Wang, Journal of Analytical and Applied Pyrolysis, 2018, 134, 526-535. (doi: 10.1016/j.jaap.2018.07.020)

4- Summary need to be rewritten in more scientific way.

Thank you very much for your kind comment. We have rewritten the summary part to make it scientific. And the content of the summary part was supplemented according to the modification of the results and discussion part.

5- What is the heating rate is it 20000 or 20C/s

The heating rate is 20000°C/s. I have confirmed this data by reviewing the program. Besides, some reports have mentioned this heating rate of pyrolysis apparatus in their researches. I have cited these literatures to illustrate the correctness of this data.

References:

Y. Qiao, B. Wang, P. Zong, Y. Tian, F. Xu, D. Li, F. Li and Y. Tian, Energy Conversion and Management, 2019, 199. (doi: 10.1016/j.enconman.2019.111964)

K. Ding, Z. Zhong, J. Wang, B. Zhang, L. Fan, S. Liu, Y. Wang, Y. Liu, D. Zhong, P. Chen and R. Ruan, Bioresour Technol, 2018, 261, 86-92. (doi: 10.1016/j.biortech.2018.03.138)

6- Are the reported results (not including the characterization results) in your paper reproducible? Are those come from a single experiment since the number of runs are not clearly indicated.

In order to demonstrate the repeatability of the experimental results, the experiments were carried out for three times in the past, and the experimental results were basically the same. In the original draft, we selected the data with the most obvious trend to illustrate the results. In the revised draft we integrated the results of three experiments, took the average value and added error bars to each group of data to explain the experimental results, so as to increase the credibility of the experimental results.

7- The discussion seems to be superficial, therefore more in-deep discussion is required to improve the scientific aspect. Conclusions. Some critical guidance in the evaluation of the present state and in the further study is given in this section.

Thank you very much for your kind comment.

In the discussion part of experimental results, we analyzed the specific components of the pyrolysis products and found that the catalysis was mainly the conversion of alcohols into hydrocarbons. At the same time, through the comparison of specific product content, the difference of catalytic performance between the catalyst prepared in this experiment and the ordinary molecular sieve catalyst was found. In addition, by referring to other studies, it is shown that the catalytic mechanism is the reduction of energy barrier. Meanwhile, through the comparison of products and the results in other literatures, it is speculated that alcohols are mainly deoxidized, hydrogenated and dehydrated in the process of catalysis, and the reaction pathways of some substances are described. Finally, high nitrogen content is the problem of algae pyrolytic oil. The composite molecular sieve has no effect on nitrogen content although its deoxidation effect is obvious. How to reduce the nitrogen content in the pyrolysis products of algae and its combined with composite molecular sieves to improve the quality of algae pyrolysis products can be one of the later research directions.

References:

M. MacLean, J. Nadeau, T. Gurnea, C. Tittiger and G. J. Blomquist, Insect Biochem Mol Biol, 2018, 102, 11-20. (doi: 10.1016/j.ibmb.2018.09.005)

D. Jiang, Z. Xia, S. Wang, H. Li, X. Gong, C. Yuan, A. El-Fatah Abomohra, B. Cao, X. Hu, Z. He and Q. Wang, Journal of Analytical and Applied Pyrolysis, 2019, 143. (doi: 10.1016/j.jaap.2019.104680)

Z. Li, E. Jiang, X. Xu, Y. Sun and R. Tu, Renewable Energy, 2020, 146, 1991-2007. (doi: 10.1016/j.renene.2019.08.012)

Y. Jiao, X. Fan, M. Perdjon, Z. Yang and J. Zhang, Applied Catalysis A: General, 2017, 545, 104-112. (doi: 10.1016/j.apcata.2017.07.036)

Research on catalytic pyrolysis of algae based on Py-GC /MS

Hao Zhao¹, Zhaoping Zhong^{*1}, Zhaoying Li¹ and Wei Wang¹

Key Laboratory of Energy Thermal Conversion and Control of Ministry of Education, School of Energy and Environment, Southeast University, Nanjing 210096, China.

Keywords: algae biomass; Py-GC/MS; pyrolysis; catalyst; HZSM-5/MCM-41 composite molecular sieve

1. Summary

In order to improve the quality of catalysis products of algae, composite molecular sieve catalyst was prepared by digestion and crystallization of HZSM-5 to reduce the oxygen content of the catalytic products. According to the analysis of the pyrolysis products, the best preparation conditions were chosen of TPAOH solution 2.0mol/L, CTAB solution 10wt%, crystallization temperature 110°C, digestion-crystallization time: 24-24h. The results indicate that the main function of catalysts is to promote the conversion of alcohols into hydrocarbons by reducing energy barriers. Catalyzed by the composite molecular sieve, the content of alcohols in the pyrolysis products decreased from more than 30% to less than 10%, the content of hydrocarbons increased from 20% to nearly 60%, while all the adverse components remained at a low level, which indicates that the catalytic pyrolysis products are of high quality. The great deoxidation effect of composite molecular sieves is not only due to the expansion of the range of organic matter during re-pyrolysis, but also the increasing of the residence time of pyrolysis products inside the structure for the external mesoporous structure.

2. Introduction

The global warming caused by fossil energy [1, 2] has attracted worldwide attention, and as non-renewable energy, it is difficult for fossil energy to support the sustainable development of society in the future. The development of renewable energy has become a worldwide trend, such as solar, wind, tidal and biomass energy. According to relevant researches [3], biomass energy, a kind of clean energy, plays an important role in alleviating environment pollution problems. The utilization of biomass could be realized by means of biomass energy conversion technology which mainly includes liquefaction, gasification and pyrolysis. Pyrolysis [4] is considered as a promising technology, which refers to the technology that organic materials decomposed into solid, liquid and gas products (bio-char, bio-oil, non-condensable

*Author for correspondence (zzhong@seu.edu.cn).

†Present address: Key Laboratory of Energy Thermal Conversion and Control of Ministry of Education, School of Energy and Environment, Southeast University, Nanjing 210096, China.

gas) at high temperature (300–1000°C) in an inert atmosphere. Nowadays researches on biomass pyrolysis mainly focus on wood biomass, while few on aquatic biomass. Take algae as example, algae belong to low-grade, oxygen-releasing autotrophic plants, with varieties of species and wide distribution. Moreover, the output of many kinds of algae in China ranks first in the world. Most algae belong to single-celled organisms [5], which means it will be easy to be improved, and can be cultivated by changing the cultivation conditions to produce species more suitable for pyrolysis [6, 7]. Compared with the first generation of biofuels [8] (edible biomass, sugar and starch plants) and the second generation of biomass [9] (lignocellulosic biomass), algae has several prominent advantages [10]: (1) High photosynthetic efficiency, which is conducive to alleviate the greenhouse effect problem; (2) Nutrients (N,P) can be extracted from waste-water and returned to the soil by waste product of fertilization; (3) Algae has a short breeding cycle [13], and its process of breeding is easy to realize automation. (4) Algae does not need to occupy arable lands, and is less affected by seasons and regions. According to the analysis composition of algae, algae has high lipid accumulation, which is suitable for the decomposition of bio-oil by heat conversion technology. Considering above advantages, algae is a potential biomass material.

There are always problems with biological oils prepared by direct pyrolysis of biological substances, such as low calorific value, high acid content and low hydrocarbon content. Therefore, measures should be taken to improve the quality of biological oils. One of common methods is to use catalysts, such as molecular sieve, metals and metal oxides. Among them, metal oxide has large pore diameter, strong water stability, and certain deoxidation performance, which is helpful to improve the stability of biological oil; Alkali metals mainly include sodium salt, potassium salt, calcium salt and their oxides; Microporous molecular sieve refers to molecular sieve with pore diameter less than 2nm, which has good deoxidation and aromatization properties. HZSM-5 [17–19], one common molecular sieve, has a microporous structure that allows pyrolysis steam to enter the interior for further pyrolysis. Under the catalysis of HZSM-5, the release of oxygen-containing gas (CO and CO₂) shows a significant decline, and the conversion of furans to aromatic hydrocarbons may be promoted over strong acid site [17, 18]. However, due to its small pore size [20], the yield of water and gas increased while the yield of organic matter decreased obviously. Besides, catalysis of HZSM-5 may be deactivated due to the polymerization of a mass of oxycompound [21]. The mesoporous zeolite MCM-41 with a larger pore size provides larger surface area and more accessible reaction sites. It has been pointed that the mild acidity of MCM-41 catalyst provides an ideal environment for controlling the conversion of high molecular weight lignocellulosic molecules [18, 20]. However, products will escape before complete re-pyrolysis because the pore size is too large [18]. Therefore, the utilization of fracture properties of macroporous catalysts and the re-framing properties of microporous molecular sieve catalysts as catalysts for biomass catalytic pyrolysis has received extensive attention [18]. In order to improve the quality of bio-oil from algae, existing studies have shown that oxygen content and acid compound of biological oil can be reduced when catalyzed by Ni supported zeolites [11], while nitrogen content of bio-oil can be reduced when catalyzed by Mg-Al layered double oxide/ZSM-5 composites with a Mg/Al molar ratio

of four (MgAl₄-LDO/ZSM-5) [12]. Hydrothermally carbonized [13] can increase the maximum weight loss rate of algae during pyrolysis and can be combined with catalytic pyrolysis to realize the reduction of nitrogen content in biological oil; ZSM-5 catalytic co-pyrolysis [14] can be a favorable process to enhance the yield of upgraded bio-oil. At present, composite molecular sieve catalyst is widely used in lignin biomass [15, 16] and shows relatively superior catalytic performance. Previous studies have used the addition of macroporous and mesoporous molecular sieves to LOSA-1 to improve the selectivity of low carbon olefins and aromatic hydrocarbons [18]. Meanwhile, the mixture of HZSM-5 and MCM-41 has been studied to improve the catalytic pyrolysis effect of fresh straw [22]. However, few researches have conducted on the composite of mesoporous and microporous molecular sieve and its application on proteinaceous biomass.

In this study, a hierarchical micro-mesoporous composite molecular sieve HZSM-5/MCM-41 with external mesoporous and internal microporous was developed through digestion and reassembly of molecular sieve HZSM-5, so as to meet re-pyrolysis requirements of pyrolysis steam in a wider range and improve the quality of pyrolysis products. With the support of Py-GC/MS [23] (pyrolysis-gas chromatography/mass spectrometry), this study intends to explore the catalytic pyrolysis products of algae and find suitable catalysts for catalytic pyrolysis to improve the quality of pyrolysis oil.

3. Materials and Methods

3.1 Experimental materials

The algae used in this study was spirulina, the selected molecular sieve catalyst was HZSM-5 (SiO₂/Al₂O₃ ratio: 26 [24]), and the ratio of biomass to catalyst was 1:2 [18, 23]. Pyrolysis products were determined by Py-GC/MS [23]. The main characteristics of spirulina can be seen in Table 1. The content of oxygen was calculated from the difference of 100% and the mass ratio of C, H, N, S.

Table 1. The main characteristics of spirulina

Feedback	Proximate analysis (Wt%) ^{ad}				Ultimate analysis (Wt%) ^{ad}				
	A	V	FC	M	C	H	O*	N	S
spirulina	5.4	73.7	12.5	7.7	45.7	7.1	35.7	10.7	0.8

Note: ad denotes air-dry basis; * was calculated by difference; A means ash; V means volatiles; FC means fixed carbon; M means moisture.

TPAOH (Tetra propylammonium hydroxide) was dissolved into 50ml ultrapure water with a certain mass fraction to prepare TPAOH solution, and then 10g HZSM-5 was added, stirred and heated in water bath at 40°C for 1h. Next CTAB (Cetyltrimethylammonium Bromide) solution of a certain concentration was mixed with HZSM-5 solution and heated in the water bath at 26°C for 1h. Then, it was poured into the reactor for 24h at a certain temperature for resolution. After adjusting the pH of the solution to 8.50 [2] by dilute sulfuric acid solution (5%), it was poured into the reactor for 24h for crystallization. After the solution was filtered and dried, dried samples were heated at 550°C for 6h, after which the preparation of catalyst composite molecular sieve HZSM-5/MCM-41 was completed.

3.2 Pyrolysis experiment

According to relevant studies, the temperature of pyrolysis by Pyroprobe 5200 pyrolyzer [35] (CDS Analytical) in this study was 600°C [25, 26] and the pyrolysis time was 20s. Pour 0.5mg catalyst, 0.5mg biomass raw material and 0.5mg catalyst into the quartz capillary tube successively, and use quartz wool to separate each layer and seal both ends. The heating rate was set at 20,000°C/s [36], and the tube was kept at the reaction temperature for 20s. High-purity helium (99.999%, Nanjing Maikesi Nanfen Special Gas Co., Ltd.) was used as pyrolysis and carrier gas, with a constant flow rate of 1.0 mL/min. Products were identified by comparison with the NIST MS library database, the analysis results were compared with the references to increase the reliability [15, 34, 36]. A DB-5 mas' capillary column (0.25 mm × 0.25 μm × 30 m) was used for product separation. The temperature in the oven was programmed to increase from 50°C (kept for 1 min) to 290°C (kept for 2 min) at a heating rate of 8°C/min. Pyrolysis of biomass raw materials, catalytic pyrolysis by HZSM-5 and composite molecular sieve HZSM-5/MCM-41 prepared under various preparation conditions were conducted respectively. The analysis of organic composition was performed on a 7890A/5975C GC/MS analyzer (Agilent Technologies).

3.3 Relative content

In the present work, the relative content (R_{content}) of organic pyrolytic products are calculated using a semi-quantitative method based on the area percentages of the chromatograph peaks and defined as follows [19, 27]:

$$R_{\text{content}} = \frac{P_i}{P_{\text{total}}}$$

where P_i is the peak area of a specific identified product, and P_{total} is the total peak area under a certain condition.

4. Results and discussion

4.1 Influence of concentration of TPAOH on catalytic effect

In the preparation of catalyst, TPAOH solution is used to dissolve the crystal nucleus of HZSM-5, so that enough aluminosilicate is dissolved into the solution, preparing for the assembly of MCM-41 on the surface of HZSM-5 in the later stage. At the same time, digestion increases the specific surface area of HZSM-5, which is conducive to the

formation of structure of mesoporous on the surface. The concentration of TPAOH solution has an impact on the degree of digestion and affects the concentration of aluminosilicate in the solution and the specific surface area of HZSM-5 as a result.

Fig.1 Distribution of pyrolysis products under catalysts with different TPAOH concentrations

In this study, concentrations of TPAOH solution in the preparation of composite molecular sieve HZSM-5/MCM-41s were selected as 1.0, 1.5, 2.0, 2.5 and 3.0 mol/L respectively. As can be seen from Fig.1, the change of the distribution of pyrolysis products follows the change of the concentration of TPAOH solution. As to the content of hydrocarbon, when the concentration of TPAOH is less than 2.0 mol/L, the content of hydrocarbon improves with the increase of the concentration and reaches a maximum of 59.17% at 2.0 mol/L. It may result from that the increase of concentration improves the concentration of aluminosilicate in the solution and the specific surface area of HZSM-5, which is conducive to the formation of composite molecular sieve HZSM-5/MCM-41. While the concentration of hydrocarbon products continues to increase, the content of hydrocarbon products declines. As to that of alcohol, when the concentration of TPAOH solution is 1.5 mol/L, 2.0 mol/L, 2.5 mol/L, the contents of alcohol are roughly the same of 7.5%. But when the concentration is 1.0 mol/L or 3.0 mol/L, the content of alcohol can be found an apparent increase, reach 13%. The reason for this phenomenon may be that the microporous structure of HZSM-5 begins to collapse with the increase of the concentration of TPAOH and it is difficult to form an appropriate aperture for the complete re-pyrolysis of pyrolysis gas [33]. For other substances, the change of the concentration of TPAOH does not cause significant variation on their content.

4.2 Influence of the concentration of CTAB on catalytic effect

In the preparation process of composite molecular sieve HZSM-5/MCM-41, CTAB solution was used as template to establish the structure of mesoporous on the surface of HZSM-5 during the crystallization process. The concentration of CTAB solution affects its interaction

with aluminosilicate during the crystallization process and thereby influence the formation of MCM-41 outside HZSM-5.

Fig.2 Distribution of pyrolysis products under catalysts with different CTAB concentrations

The concentration of CTAB solution chosen in the preparation of composite molecular sieve HZSM-5/MCM-41s in this study are 5, 10, 15, 20wt% respectively. As exhibited in Fig. 2, when the concentration of CTAB is below 10wt%, the content of hydrocarbons in the pyrolysis product increases along with the increase of the concentration of CTAB in the solution. For this phenomenon, it may due to that the increasing CTAB solution concentration makes for the formation of the orderly six-party structure and as a result form a better structure of composite molecular sieve HZSM-5/MCM-41 after demolded. However, when the concentration of CTAB keeps growing, a decline can be seen in the content of hydrocarbons in the pyrolysis products. It is possible that the hexagonal structure starts to become disordered, and the damage to the microporous structure of HZSM-5 is excessive [28]. When the concentration of CTAB is 10wt%, the damage to microporous structures of HZSM-5 is relatively weaker, and the MCM-41 structure can be assembled on the surface. Therefore, its catalytic effect is the best, with the highest content of hydrocarbon of 59.17% and the lowest content of acid substances of 2.79%. At the same time the content of aldehydes and ketones is low, which ensures the stability of pyrolysis products effectively.

4.3 Influence of different temperatures of crystallization on catalytic effect

The temperature of resolution during the preparation of the molecular sieve was 110 °C, and the temperature of crystallization were 90 °C, 100 °C, 110 °C and 120 °C.

Fig. 3 Distribution of catalyst products at different crystallization temperatures

As the Fig. 3 shows, when the crystallization temperature is lower than 110 °C, the increase of temperature promoted the increase of the content of hydrocarbons in the pyrolysis products. It may be due to that the increase of temperature promotes the reaction between dissolved crystal nucleus and template agent, which is conducive to the assembly of MCM-41 structures on the external surface of HZSM-5 and the formation of composite molecular sieve structures. However, with the temperature keeps increasing, decline can be seen in the content of hydrocarbon and increase can be found in the content of acids and alcohols. The explanation for this phenomenon may be that when the temperature further increases, the reaction between template ion and silicate root is unduly violent, which promotes the formation of amorphous material and destroys the assembly of mesoporous structures [27]. When the crystallization temperature is 110 °C, the content of hydrocarbon reaches the highest (59.17%), while the contents of acids (2.79%), aldehydes (1.08%) and ketones (1.63%) reaches the lowest, which indicates that the deoxidation reaction is the most complete, and under this temperature of crystallization, the composite molecular sieve can have the most suitable structure for pyrolysis.

4.4 Influence of different digestion - crystallization time on catalytic effect

Digestion and crystallization are competing processes. The purpose of digestion is to digest HZSM-5 crystal nucleus under alkaline environment, and to improve its specific surface area to facilitate mesoporous structure assembly on its surface. At the same time, parts of the silicic acid radical dissolve into crystal fluid from crystal nucleus as a source of silicic acid radical to participate in the synthesis of MCM-41. The role of crystallization lies in the interaction between template molecule and silicon and aluminium source to form MCM-41.

Fig.4 Distribution of catalyst products under different digestion-crystallization time

In this study, the digestion-crystallization time was respectively adjusted to 12-36h, 24-24h, 36-12h. As can be seen from Fig.4, when the digestion time was 12h and the crystallization time was 36h, due to the short digestion time, the dissolved sialic acid radical is few, which hindered the assembly of MCM-41 in the next step [27]. However, when the digestion time is 36h and the crystallization time is 12h, although a large amount of sialic acid radical is dissolved, the crystallization time is too short to fully assemble the mesoporous structure, and the mesoporous structure is poor in order. When the digestion time and crystallization time are both 24h, the mesoporous structure is formed and the microporous structure is less destroyed, and the ratio between the two competitive processes of digestion and crystallization is relatively appropriate. As can be seen from Fig. 4, when the digestion-crystallization time is 24-24h, the hydrocarbon content reaches the highest level, and acids, aldehydes and ketones all reach the lowest level, indicating that the composite molecular sieve HZSM-5/MCM-41 catalyst has a great catalytic pyrolysis effect.

4.5 Comparison of pyrolysis and catalytic pyrolysis products

In this study, direct pyrolysis of algae and catalytic pyrolysis by HZSM-5 and composite molecular sieve HZSM-5/MCM-41 were conducted. The distribution of pyrolysis products is shown in Fig.5.

In the term of direct pyrolysis, due to the high content of nitrogen in algae, the content of nitrogen-containing organic compounds is significantly higher than that of herbaceous biomass, which is consistent with the previous research [25]. Due to the low lignin content of algae, the phenolic substances in their pyrolysis products are also few, accounting for only 3%. Hydrocarbon content is relatively high, up to 25%, and so as to alcohols, up to 30%, which may be due to the high axunge content of algae themselves. At the same time, contents of acids and aldehydes were also lower, at 5% and 5% respectively. It can be found

that the bio-oil generated by direct pyrolysis of algae has high quality, indicating that algae is a promising biomass raw material.

Fig.5 Distribution of pyrolysis products by direct pyrolysis, catalyzed by HZSM-5 and composite molecular sieve HZSM-5/MCM-41

In this study, preparation conditions of HZSM-5/MCM-41 composite molecular sieve were determined as: TPAOH solution: 2.0mol /L, CTAB solution: 10wt%, crystallization temperature: 110°C, digestion-crystallization time: 24-24h. Compared with the catalytic pyrolysis by HZSM-5, for the micro-pore in HZSM-5, pyrolysis products were deoxidized in re-pyrolysis, the hydrocarbon content in the products is significantly increased, alcohols are significantly reduced, and aldehydes are also reduced to some extent. Catalyzed by HZSM-5, acid substances increase slightly, from 4% to 5%; The hydrocarbon content increase significantly from 15% to 28%; Alcohols decrease significantly from 35% to 20%; The content of aldehydes also drop, from 5% to 2%; Esters also fall, from 5% to 3%. It can be considered that through the catalytic pyrolysis of HZSM-5, due to the appropriate pore size of HZSM-5, parts of oxygen-containing organic compounds further undergoes deoxidation during re-pyrolysis, and main components are some alcohols of macromolecules [26, 29]. The content of nitrogen-containing compounds is hardly affected, indicating that HZSM-5 could not promote the removal of nitrogen from nitrogen-containing organic compounds.

Compared with HZSM-5, when catalyzed by composite molecular sieve HZSM-5/MCM-41, the content of acid substances decreased from 5% to 2.17%; The hydrocarbon content increased significantly from 38% to 59.17%; Alcohols were further reduced from 20% to 8%; Aldehydes and ketones contents also declined, from 3% and 5% to 2% and 2%, respectively. It can be considered that due to the special structure of the external mesoporous and internal micropores of the composite molecular sieve HZSM-5/MCM-41, more oxygen-containing organic substances undergo deoxidation reaction during further pyrolysis inside the catalyst, while alcohols are also the main organic constituents for re-pyrolysis. Similarly,

the content of nitrogenous organic compounds is not affected, indicating that the removal of nitrogen could not be achieved by just adjusting the aperture.

Table 2. Content of alcohols and hydrocarbons in the products of direct pyrolysis, catalytic pyrolysis by HZSM-5 and HZSM-5/MCM-41 of algae

Compound	Direct (%)	HZSM-5 (%)	HZSM-5/MCM-41 (%)
Hydrocarbons			
Pentane	2.15	0.00	0.00
Cyclopropane, 1,2-dimethyl-, cis-	0.00	5.55	9.30
2-Hexene, (Z)-	0.00	0.00	2.92
2,4-Hexadiene, (Z, Z)-	0.00	0.00	1.21
3-Vinyl-1-cyclobutene	0.66	0.00	0.00
Benzene	0.00	1.27	2.95
3-Hexene, 2-methyl-, (Z)-	0.00	0.39	0.00
2-Heptene	0.00	0.47	0.00
2-Hexene, 5-methyl-, (E)-	0.83	0.52	0.00
1,4-Hexadiene, 2-methyl-	0.00	0.00	1.62
Toluene	5.82	5.70	5.47
3-Undecene, (Z)-	0.00	0.76	0.00
Ethylbenzene	1.06	1.15	1.35
p-Xylene	0.00	3.12	4.52
Styrene	0.00	0.00	0.90
1,3,5,7-Cyclooctatetraene	0.87	0.90	0.00
Benzene, propyl-	0.00	0.00	0.47
Benzene, 1-ethyl-3-methyl-	0.00	0.88	0.00
Benzene, 1-ethyl-4-methyl-	0.00	0.00	2.08
Benzene, 1,3,5-trimethyl-	0.00	0.00	0.40
D-Limonene	0.69	0.23	0.23
1,3,5-Cycloheptatriene, 7-ethyl-	0.00	0.05	0.00
Benzene, 1-ethenyl-2-methyl-	0.00	0.00	0.44
Indene	0.00	0.00	0.60
Benzene, 1,2-diethyl-	0.00	0.00	1.56
1-Phenyl-1-butene	0.00	0.00	0.33
3a,6-Methano-3aH-indene, 2,3,6,7-tetrahydro-	0.00	0.00	0.48
1H-Indene, 3-methyl-	0.00	0.00	0.38
Naphthalene	0.00	0.00	0.95
Bicyclo[4.4.1]undeca-1,3,5,7,9-pentaene	0.00	0.37	0.78
Bicyclopentyl-1,1'-diene	0.00	1.05	0.00
Tricyclo[6.4.0.0(3,7)]dodeca-1,9,11-triene	0.00	0.00	0.68
Alcohols			
Cyclopropaneethanol	6.81	0.00	0.00
2-Octyn-1-ol	0.00	0.00	0.35
trans-2-Ethyl-2-hexen-1-ol	0.00	0.00	0.33
3-Octyn-1-ol	0.00	1.97	0.00
4-Penten-1-ol, 2-methylene-	0.00	0.88	0.00
3-Pentyn-1-ol	1.04	0.40	0.00
3-Decyn-2-ol	0.00	0.00	0.34

1-Cyclohexene-1-methanol	0.00	0.72	0.00
2,5-Cyclooctadien-1-ol	0.00	0.00	0.37
2-Decyn-1-ol	0.00	0.06	0.17
2-Undecanethiol, 2-methyl-	0.00	0.13	0.00
4-Cyclopentene-1,3-diol, trans-	1.63	0.00	0.00
4-Penten-1-ol, 2-methylene-	1.16	0.00	0.00
5-Hexen-2-ol	0.42	0.00	0.00
3-Buten-2-ol, 3-methyl-	0.90	0.00	0.00
3-Hexyne-2,5-diol	0.90	0.00	0.00
2-Nonen-1-ol, (E)-	0.26	0.00	0.00
Bicyclo[2.1.1]hexan-2-ol, 2-ethenyl-	0.61	0.00	0.00
Cyclohexanol, 2-methyl-5-(1-methylethenyl)-	0.08	0.00	0.00
Isopinocarveol	0.27	0.38	0.00
2-Nitrohept-2-en-1-ol	0.19	0.00	0.00
cis-p-Mentha-2,8-dien-1-ol	0.85	0.00	0.00
1-(4-Methoxyphenyl)-2-(4-pyridaziny) ethanol	0.00	0.43	0.00
2-Naphthol, 1,2,3,4,4a,5,6,7-octahydro-4a-methyl-	0.59	0.37	0.00
5,7-Dodecadiyn-1,12-diol	0.23	0.29	0.40
1,2,4-Metheno-1H-cyclobuta[cd]pentalene-3,5-diol, octahydro-	0.00	0.00	0.45
(1-Benzyl-cyclopropyl)-methanol	0.00	0.00	0.97
Falcarinol	0.28	0.48	0.51
Ethanol, 2-(9-octadecenyloxy)-, (Z)-	3.29	3.19	1.02
Estra-1,3,5(10)-trien-17 α ol	0.00	1.20	0.28

The contents of alcohols and hydrocarbons in direct pyrolysis and catalytic pyrolysis products are shown in Table 2. Compared with direct pyrolysis products, micromolecular alcohol content in pyrolysis products catalyzed by HZSM-5 shows a significant decline, for example, the content of cyclopropane ethanol, 3-Pentyn-1-ol and 4-Cyclopentene-1,3-diol, trans- decreases respectively from 6.8%, 1.04% and 1.63% to 0, 0.4% and 0, which indicates that sufficient hydrogenation, deoxidization, dehydration reaction happens to these compounds. The addition of HZSM-5 helps to reduce the energy barrier due to hydrogen bond and van der Waals force, thus promoting the occurrence of repyrolysis. And the possible pathways [30,31] of their hydrodeoxidation reactions are show in Fig.6. As to macromolecular alcohols such as Ethanol, 2-(9-octadecenyloxy)-, (Z)-, because of the limitation of the aperture of microporous structure, HZSM-5 failed to show great catalytic pyrolysis performance. HZSM-5/MCM-41 shows great catalytic properties on micromolecular alcohols due to its internal microporous structure and strong acid sites. It also has hydrodeoxidation effect on macromolecular alcohols for its external mesoporous structure. For hydrocarbon products, compared with HZSM-5, the content of each product increased under the catalysis of HZSM-5/MCM-41, which could be attributed to the fact that the crust-core structure of HZSM-5/MCM-41 increased the residence time of pyrolysis gas in the structure and thus realize more sufficient hydrodeoxidation performance.

Compared with bio-char, Ni/SBA-15 [12] and activated carbon [13], HZSM-5/MCM-41 demonstrated excellent deoxidation performance on alcohols; Compared with composite molecular sieve prepared with NaOH [34], the reaction of digestion using TPAOH is mild and easy to be controlled. Previous studies involving catalysts modified by an alkali treatment have focused on alkali-treated HZSM-5 at low concentration ($<1.0\text{mol/L}$) [34] of inorganic base. The high concentration (2.0mol/L) of TPAOH solution is more effective for HZSM-5 treatment than the low concentration base solution ($< 1.0\text{mol/L}$). Moreover, the modified catalyst does not require NH_4NO_3 -ion exchange. The 2.0mol/L TPAOH modified HZSM-5 led to produce the highest aromatic selectivity and catalytic activity from CFP of algae.

Fig.7 The possible hydrodeoxygenation pathways of partial alcohols

4.6 Microstructure of HZSM-5 and HZSM-5/MCM-41

In order to study the effect of digestion and crystallization on the microstructure of HZSM-5, SEM characterization of catalyst samples was performed. Fig.7(a) is the SEM figure of molecular sieve HZSM-5. It can be seen from the figure that the HZSM-5 is of regular cubic shape with smooth surface. Fig.7(b) shows the SEM figure of HZSM-5/MCM-41 composite molecular sieve HZSM-5/MCM-41. It can be seen that the surface becomes fragmented and irregular, and many small particles are generated to adhere to the surface of crystal nucleus, namely, the synthetic MCM-41. As can be seen from the figure, compared with the molecular sieve of HZSM-5/MCM-41, the surface pore diameter of the composite molecular sieve HZSM-5/MCM-41 of HZSM-5 is significantly increased, and at the same time, it can maintain a regular overall structure without excessive collapse of the skeleton, which may be the reason for its better catalytic deoxygenation effect

Fig.7(a) SEM of HZSM-5

Fig.7(b) SEM of HZSM-5/MCM-41

5. Conclusion

By investigating the preparation conditions of HZSM-5/MCM-41, it is determined that under the condition of TPAOH concentration 2.0 mol/L, CTAB concentration 10 wt%, crystallization temperature 110 °C, digestion-crystallization time 24-24 h, the external mesoporous structure of the composite molecular sieve can be sufficiently formed and the internal microporous structure would not be excessively damaged. The main components in the direct pyrolysis products of algae are alcohols, followed by N-containing compounds and hydrocarbons, while the adverse components aldehydes, ketones and acids have low contents, which indicates the bio-oil are of high quality. Compared with direct pyrolysis, catalytic pyrolysis by HZSM-5 can effectively reduce the oxygen content of pyrolysis products, mainly by reducing the content of alcohols. However, the adverse components do not change significantly due to their low content, and the content of N-containing compounds is hardly

affected. After catalytic pyrolysis by HZSM-5/MCM-41, the alcohols decrease significantly again, and the content of adverse components also decrease, but the content of nitrogen-containing organic matter is not affected. After analysing the specific substances, it is found that HZSM-5 had great hydrodeoxygenation performance on micromolecular alcohols, but the catalytic effect was limited by its pore size. The external mesoporous structure of HZSM-5 / MCM-41 not only widens the reaction range of hydrodeoxygenation reaction, but also increases the residence time of pyrolysis products in the internal structure, making deoxygenation hydrogenation reaction more complete. Through the analysis of the products, it is found that the alcohols are mainly converted into hydrocarbons by hydrogenation, deoxygenation and dehydration under the catalysis. Finally, high nitrogen content is the problem of algae pyrolytic oil. The composite molecular sieve has no effect on nitrogen content although its deoxygenation effect is obvious. How to reduce the nitrogen content in the pyrolysis products of algae and its combined with composite molecular sieves to improve the quality of algae pyrolysis products can be one of the later research directions.

Funding Statement

This work was sponsored by the National Natural Science Foundation of China (No.51776042).

Data Accessibility

Electronic supplementary information (ESI) data are available within the Dryad Digital Repository: <https://doi.org/10.5061/dryad.22829q7>

Reviewer URL:

<https://datadryad.org/stash/share/Y8cvfb4khkDnyVhByN5VdGlzJwzBEJGPOzcDDC8X5PY>

Competing Interests

We have no competing interests.

Acknowledgements

Authors' Contributions

H.Z. analysed the results and drafted the manuscript. ZY.L. collected the samples, recorded the spectra and processed the spectra tighter with H.Z. W.W. participated in all the tests. ZY.L. helped with interpretation of the data. ZP.Z. revised the manuscript. W.W. reviewed the draft.

References

1. K. E. Lonngren and E.-W. Bai, *Energy Policy*, 2008, **36**, 1567-1568. (doi: [10.1016/j.enpol.2007.12.019](https://doi.org/10.1016/j.enpol.2007.12.019))
2. B. M. E. Chagas, C. Dorado, M. J. Serapiglia, C. A. Mullen, A. A. Boateng, M. A. F. Melo and C. H. Ataíde, *Fuel*, 2016, **179**, 124-134. (doi: [10.1016/j.fuel.2016.03.076](https://doi.org/10.1016/j.fuel.2016.03.076))
3. Danish and Z. Wang, *Sci Total Environ*, 2019, **670**, 1075-1083. (doi: [10.1016/j.scitotenv.2019.03.268](https://doi.org/10.1016/j.scitotenv.2019.03.268))
4. B. Cao, S. Wang, Y. Hu, A. E.-F. Abomohra, L. Qian, Z. He, Q. Wang, B. B. Uzojinwa and S. Esakkimuthu, *Renewable Energy*, 2019, **138**, 29-38. (doi: [10.1016/j.renene.2019.01.084](https://doi.org/10.1016/j.renene.2019.01.084))
5. A. Singh, P. S. Nigam and J. D. Murphy, *Bioresour Technol*, 2011, **102**, 10-16. (doi: [10.1016/j.biortech.2010.06.032](https://doi.org/10.1016/j.biortech.2010.06.032))
6. S. Taghavi, O. Norouzi, A. Tavasoli, F. Di Maria, M. Signoretto, F. Menegazzo and A. Di Michele, *International Journal of Hydrogen Energy*, 2018, **43**, 19918-19929. (doi: [10.1016/j.ijhydene.2018.09.028](https://doi.org/10.1016/j.ijhydene.2018.09.028))
7. H. Chen, Z. He, B. Zhang, H. Feng, S. Kandasamy and B. Wang, *Energy*, 2018, **158**, 103-110. (doi: [10.1016/j.energy.2018.07.078](https://doi.org/10.1016/j.energy.2018.07.078))

- 2019, 179, 1103–1113. (doi: 10.1016/j.energy.2019.04.184)
8. D. F. Correa, H. L. Beyer, H. P. Possingham, S. R. Thomas–Hall and P. M. Schenk, *Renewable and Sustainable Energy Reviews*, 2017, 74, 1131–1146. (doi: 10.1016/j.rser.2017.02.068)
9. C. Yuan, S. Wang, B. Cao, Y. Hu, A. E.-F. Abomohra, Q. Wang, L. Qian, L. Liu, X. Liu, Z. He, C. Sun, Y. Feng and B. Zhang, *Energy*, 2019, 173, 413–422 (doi: 10.1016/j.energy.2019.02.091)
10. I. S. Tan, M. K. Lam, H. C. Y. Foo, S. Lim and K. T. Lee, *Chinese Journal of Chemical Engineering*, 2019, (doi: 10.1016/j.cjche.2019.05.012)
11. N. H. Zainan, S. C. Srivatsa, F. Li and S. Bhattacharya, *Fuel*, 2018, 223, 12–19. (doi: 10.1016/j.fuel.2018.02.166)
12. L. Gao, J. Sun, W. Xu and G. Xiao, *Bioresour Technol*, 2017, 225, 293–298. (doi: 10.1016/j.biortech.2016.11.077)
13. E. Ansah, L. Wang, B. Zhang and A. Shahbazi, *Fuel*, 2018, 228, 234–242. (doi: 10.1016/j.fuel.2018.04.163)
14. Y.-M. Kim, J. Jeong, S. Ryu, H. W. Lee, J. S. Jung, M. Z. Siddiqui, S.-C. Jung, J.-K. Jeon, J. Jae and Y.-K. Park, *Energy Conversion and Management*, 2019, 195, 727–737. (doi: 10.1016/j.enconman.2019.05.034)
15. R. Zou, Y. Wang, L. Jiang, Z. Yu, Y. Zhao, Q. Wu, L. Dai, L. Ke, Y. Liu and R. Ruan, *Bioresour Technol*, 2019, 289, 121609. (doi:10.1016/j.biortech.2019.121609)
16. B. Cao, Z. Xia, S. Wang, A. E.-F. Abomohra, N. Cai, Y. Hu, C. Yuan, L. Qian, L. Liu, X. Liu, B. Li, Z. He and Q. Wang, *Journal of Analytical and Applied Pyrolysis*, 2018, 134, 526–535. (doi: 10.1016/j.jaap.2018.07.020)
17. X. Zhang, C. Li, A. Tian, Q. Guo and K. Huang, *Carbon Resources Conversion*, 2019, 2, 111–116. (doi: 10.1016/j.crcon.2019.05.002)
18. H. Zhang, R. Xiao, B. Jin, G. Xiao and R. Chen, *Bioresour Technol*, 2013, 140, 256–262. (doi: 10.1016/j.biortech.2013.04.094)
19. B. Zhang, G. Tan, Z. Zhong and R. Ruan, *Journal of Analytical and Applied Pyrolysis*, 2017, 123, 92–98. (doi: 10.1016/j.jaap.2016.12.022)
20. D. Wang, R. Xiao, H. Zhang and G. He, *Journal of Analytical and Applied Pyrolysis*, 2010, 89, 171–117. (doi: 10.1016/j.jaap.2019.04.014)
21. T. R. Carlson, J. Jae, Y.-C. Lin, G. A. Tompsett and G. W. Huber, *Journal of Catalysis*, 2010, 270, 110–124. (doi: 10.1016/j.jcat.2009.12.013)
22. X. Li, L. Dong, J. Zhang, C. Hu, X. Zhang, Y. Cai and S. Shao, *Journal of the Energy Institute*, 2019, 92, 136–143. (doi: 10.1016/j.joei.2017.10.015)
23. Y. Jiang, P. Zong, B. Tian, F. Xu, Y. Tian, Y. Qiao and J. Zhang, *Energy Conversion and Management*, 2019, 179, 72–80. (doi: 10.1016/j.enconman.2018.10.049)
24. H. Li, S. He, K. Ma, Q. Wu, Q. Jiao and K. Sun, *Applied Catalysis A: General*, 2013, 450, 152–159. (doi: 10.1016/j.apcata.2012.10.014)
25. C. Lorenzetti, R. Conti, D. Fabbri and J. Yanik, *Fuel*, 2016, 166, 446–452. (doi: 10.1016/j.fuel.2015.10.051)
26. Z. Yu, M. Dai, M. Huang, S. Fang, J. Xu, Y. Lin and X. Ma, *Renewable Energy*, 2018, 125, 465–471. (doi: 10.1016/j.renene.2018.02.136)
27. Z. Li, Z. Zhong, B. Zhang, W. Wang and W. Wu, *Journal of Analytical and Applied Pyrolysis*, 2019, 138, 103–113. (doi: 10.1016/j.jaap.2018.12.013)
28. S. Abdi and D. Dorrnanian, *Optics & Laser Technology*, 2018, 108, 372–377. (doi: 10.1016/j.optlastec.2018.07.009)
29. F. J. Keil, *Microporous and Mesoporous Materials*, 1999, 29, 49–66. (doi: 10.1016/S1387-1811(98)00320-5)
30. M. MacLean, J. Nadeau, T. Gurnea, C. Tittiger and G. J. Blomquist, *Insect Biochem Mol Biol*, 2018, 102, 11–20. (doi: 10.1016/j.ibmb.2018.09.005)
31. D. Jiang, Z. Xia, S. Wang, H. Li, X. Gong, C. Yuan, A. El-Fatah Abomohra, B. Cao, X. Hu, Z. He and Q. Wang, *Journal of Analytical and Applied Pyrolysis*, 2019, 143. (doi: 10.1016/j.jaap.2019.104680)
32. Z. Li, E. Jiang, X. Xu, Y. Sun and R. Tu, *Renewable Energy*, 2020, 146, 1991–2007. (doi: 10.1016/j.renene.2019.08.012)
33. Y. Jiao, X. Fan, M. Perdjon, Z. Yang and J. Zhang, *Applied Catalysis A: General*, 2017, 545, 104–112. (doi: 10.1016/j.apcata.2017.07.036)
34. S. Tang, C. Zhang, X. Xue, Z. Pan, D. Wang and R. Zhang, *Journal of Analytical and Applied Pyrolysis*, 2019, 137, 86–95. (doi: 10.1016/j.jaap.2018.11.013)
35. Y. Qiao, B. Wang, P. Zong, Y. Tian, F. Xu, D. Li, F. Li and Y. Tian, *Energy Conversion and Management*, 2019, 199. (doi: 10.1016/j.enconman.2019.111964)
36. K. Ding, Z. Zhong, J. Wang, B. Zhang, L. Fan, S. Liu, Y. Wang, Y. Liu, D. Zhong, P. Chen and R. Ruan, *Bioresour Technol*, 2018, 261, 86–92. (doi: 10.1016/j.biortech.2018.03.138)

37. Zhao Hao, Zhong Zhao-P, Li algae based on Py-GC/MS, Dryad (https://doi.org/10.5061/dryad.228
Zhao-Y, Wang Wei. 2019 Data from: Digital Repository. 29q7)
Research on catalytic pyrolysis of